# Airborne aerosol olfactory deposition contributes to anosmia in COVID-19

**Alan D. Workman** [1,2]*, **Aria Jafari** [1,2], **Roy Xiao**[1,2], **Benjamin S. Bleier**[1,2]

**1** Department of Otolaryngology, Massachusetts Eye and Ear Infirmary, Boston, Massachusetts, United States of America, **2** Harvard Medical School, Boston, Massachusetts, United States of America

* Alan_Workman@meei.harvard.edu

**Data Availability Statement:** All relevant data is included in supplementary files.

**Funding:** The authors received no specific funding for this work.

## Abstract

### Introduction

Olfactory dysfunction (OD) affects a majority of COVID-19 patients, is atypical in duration and recovery, and is associated with focal opacification and inflammation of the olfactory epithelium. Given recent increased emphasis on airborne transmission of SARS-CoV-2, the purpose of the present study was to experimentally characterize aerosol dispersion within olfactory epithelium (OE) and respiratory epithelium (RE) in human subjects, to determine if small (sub 5µm) airborne aerosols selectively deposit in the OE.

### Methods

Healthy adult volunteers inhaled fluorescein-labeled nebulized 0.5–5µm airborne aerosol or atomized larger aerosolized droplets (30–100µm). Particulate deposition in the OE and RE was assessed by blue-light filter modified rigid endoscopic evaluation with subsequent image randomization, processing and quantification by a blinded reviewer.

### Results

0.5–5µm airborne aerosol deposition, as assessed by fluorescence gray value, was significantly higher in the OE than the RE bilaterally, with minimal to no deposition observed in the RE (maximum fluorescence: OE 19.5(IQR 22.5), RE 1(IQR 3.2), p<0.001; average fluorescence: OE 2.3(IQR 4.5), RE 0.1(IQR 0.2), p<0.01). Conversely, larger 30–100µm aerosolized droplet deposition was significantly greater in the RE than the OE (maximum fluorescence: OE 13(IQR 14.3), RE 38(IQR 45.5), p<0.01; average fluorescence: OE 1.9 (IQR 2.1), RE 5.9(IQR 5.9), p<0.01).

### Conclusions

Our data experimentally confirm that despite bypassing the majority of the upper airway, small-sized (0.5–5µm) airborne aerosols differentially deposit in significant concentrations within the olfactory epithelium. This provides a compelling aerodynamic mechanism to explain atypical OD in COVID-19.

**Competing interests:** The authors do not have competing interests that could bias this work.

## Introduction

COVID-19 represents an extraordinary global health threat with multi-organ sequelae. Olfactory dysfunction (OD) has emerged in the majority of cases and is predictive of milder disease [1, 2]. However, OD in COVID-19 is unlike typical post-viral smell loss in that it occurs largely in the absence of other upper airway complaints [1, 3]. This symptom pattern is consistent with radiographic findings of severe focal olfactory epithelium (OE) inflammation with otherwise normal nasal respiratory epithelium (RE, Fig 1) [4]. The explanation for this selective olfactory involvement remains mysterious as the OE has the lowest expression of angiotensin converting enzyme 2 (ACE2), the SARS-CoV-2 binding receptor, within the entire nasal cavity [5]. A recent open letter from 239 scientists to international public health organizations called for acknowledgement of airborne spread as a potential transmission mode for SARS-CoV-2 [6] with the subsequent recognition by the World Health Organization in a scientific brief [7]. Given the increased emphasis on airborne transmission, we hypothesized that small concentrations of persistently airborne aerosols (those below 5μm) would be more prone to dispersing within the olfactory epithelium than the lower nasal airway. This selective olfactory dispersion of smaller-sized airborne particulate would therefore provide a novel mechanism linking atypical olfactory dysfunction in COVID-19 with airborne transmission. The purpose of this study was to therefore experimentally characterize whether airborne aerosols are capable of selective OE dispersal within the healthy human nasal cavity.

## Methods

The Mass General Brigham IRB approved the IRB Protocol, 2020P-001246. Written informed consent was obtained from all subjects. Participants were healthy subjects between the ages of 25 and 35 without a history of chronic rhinosinusitis, allergic rhinitis, or other rhinologic disease. At the time of data collection, subjects did not have any symptoms of acute sinusitis, rhinorrhea, or subjective nasal obstruction. A Hudson RCI 1883 nebulizer (Teleflex Medical, Morrisville, NC was used to generate smaller sized particulate in the sub 5μm range, classically characterized as a particle size capable of being persistently airborne for extended periods of time and having protracted settling rates. An optical particle sizer (OPS 3330, TSI Inc, Shoreview, MN) was utilized to measure the size distribution of nebulized airborne particulates to ensure that particles in this range were generated. Alternatively, an atomizer was used to generate larger aerosolized particulate, in the 30–100 μm range (MADomizer, Teleflex, Wayne,

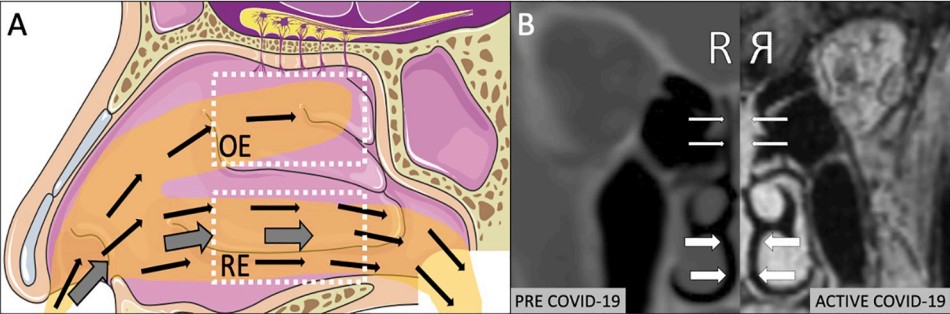

**Fig 1.** A) Illustration of classic description of airborne aerosols (thin arrows) bypassing the nasal respiratory epithelium (RE) with proposed mechanism for a proportion of these aerosols penetrating into the olfactory epithelium (OE). Thick arrows depict larger-sized aerosolized droplets settling within RE (adapted from Servier Medical Art). B) Matched right sided CT and MRI (reflected) scan of same patient demonstrating new onset isolated inflammation of OE (thin arrows) during COVID-19 infection with sparing of RE (thick arrows).

PA). A fluorescein solution of 1mg FUL-GLO Fluorescein Sodium (NDC17478-404-01, Akorn, Inc, Lake Forest, IL, USA) in 5mL saline was utilized in both conditions in the respective source of particle generation (nebulizer vs. atomizer). Immediately following 60 seconds of exposure to nebulization or atomization, subjects then underwent rigid nasal endoscopy equipped with a blue light liter for fluorescein visualization (Karl Storz, Tuttlingen, Germany). Digital images were captured of the OE (olfactory cleft, superior middle turbinate) and RE (nasal floor, inferior turbinate) bilaterally, randomized, and provided to a blinded reviewer for image processing using ImageJ (version 2.0.0-rc-69/1.52p). Images were systematically color-adjusted, thresholded, and transformed to 8-bit grayscale for quantification. Maximum fluorescence intensity, average-intensity of non-zero pixels, and standard error of non-zero pixels were calculated, with subtraction of non-fluorescein stained background values.

## Statistics

Stata version 13 (StataCorp, College Station, TX) was used for statistical analysis using the within-program Mann-Whitney U test comparing maximum and average fluorescent values between the RE and OE (within subjects, n = 3). No outlier data was excluded, and there were no missing values for any experimental replicate. Medians and interquartile range (IQR) are reported as median (IQR spread). A p-value of $<0.05$ was considered significant.

## Results

Nebulizer particulate analysis with an optical particle sizer confirmed reliable sub 5μm airborne aerosol generation with a peak of 2μm. Following fluorescein-labeled airborne aerosol inhalation of nebulized particulate, a significantly higher deposition of aerosols were found in the OE than the RE with minimal to no deposition observed in the RE (maximum fluorescence: OE 19.5(22.5), RE 1(3.2), $p<0.001$, U = 0, n = 10,6; average fluorescence: OE 2.3(4.5), RE 0.1(0.2) $p<0.01$, U = 3, n = 10,6, Mann-Whitney U test, n = 3 subjects, Fig 2).

Conversely, following fluorescein labeled aerosolized droplet (30–100 μm) inhalation, significantly greater deposition was observed in the RE as compared with the OE (maximum

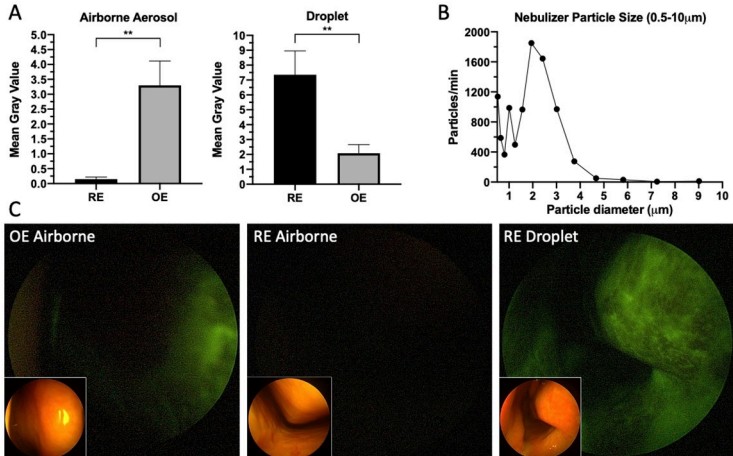

**Fig 2.** A) Small airborne aerosol (0.5–5μm) deposition versus aerosolized droplet (30–100 μm) deposition of fluorescein solution in the respiratory epithelium (RE) and olfactory epithelium (OE) (small airborne aerosol average fluorescence: OE 2.3(IQR 4.5), RE 0.1(IQR 0.2), $p<0.01$, aerosolized droplet average fluorescence: OE 1.9(IQR 2.1), RE 5.9(IQR 5.9), $p<0.01$). B) Size distribution of airborne aerosols in the 0.5–10μm range produced during nebulizer use. C) Blue-light filtered endoscopic nasal images (with brightfield insets) of fluorescein airborne and droplet labeled distribution to the olfactory (OE) and respiratory epithelium (RE) in human subjects.

fluorescence: OE 13(14.3), RE 38(45.5) p<0.01, U = 2, n = 6,6; average fluorescence: OE 1.9 (2.1), RE 5.9(5.9), p<0.01, U = 1, n = 6,6, Mann-Whitney U test, n = 3 subjects, Fig 2A).

## Discussion

Self-reported OD has been widely described in COVID-19 through a variety of case reports [4], case series [8] and surveys [9], with a pooled prevalence of 52.7% [2]. The true rate is likely higher as formal smell testing by Moein et al. [8] revealed OD in 98% of COVID-19 positive patients, only 35% of whom self-reported. Using an olfaction survey, Yan et al. [10] found that loss of smell was in fact more highly correlated with COVID-19 positivity than any other systemic or pulmonary symptom. This correlative finding is echoed in a study utilizing a symptom tracker app in the United Kingdom [9]. Yan et al further demonstrated that self-reported OD was associated with milder disease and most often occurred in the absence of other symptoms, unlike typical post-viral OD. This finding was confirmed by Kaye et al, showing unexpectedly low rates of nasal congestion (25%) and rhinorrhea (18%) in a cohort of COVID-19 positive anosmic patients [3]. These results suggest that SARS-CoV-2 exhibits a unique predilection for impacting the OE to the relative exclusion of the remainder of the nasal airway, a feature corroborated by published reports of nasal CT and MRI scans(Fig 1B) [4].

The simplest mechanism for this distinct and isolated OE inflammation would be evidence of SARS-CoV-2 tropism for either the olfactory neurons or the OE itself. The neurotropic hypothesis appears unlikely as olfactory neurons lack both the ACE2 protein and TMPRSS2 gene required for viral spike protein binding and cellular entry [5]. Furthermore the rapidity of recovery [10] and lack of protective effect among females [8], otherwise common in neuropathic OD, suggest a non-neural etiology. In contradistinction, OE tropism could be explained by the presence of ACE2 within the sustentacular cells which support the olfactory neurons [5]. However, Brann et al. [5] demonstrated that ACE2 expression was actually higher within the ciliated and secretory cells found throughout the remainder of the RE than in the OE sustentacular cells. Therefore, ACE2 avidity alone cannot explain this phenomenon.

In the absence of a biologic explanation for atypical OD, our study explored the feasibility of a mechanism of differential particulate deposition. Small-sized, persistently airborne aerosols (less than 5–10μm) are classically understood to bypass the upper airway in favor of alveolar deposition [11]. While our results confirm this effect within the RE, they also reveal that airborne aerosols in fact deposit in appreciable concentrations within the OE. This effect is unique to small-sized airborne aerosols as we found that larger aerosolized droplets (30–100μm) were more likely to deposit in the RE. Based on these distribution patterns, we therefore surmise that against the background of widely distributed ACE2 throughout both the OE and RE, low concentrations of airborne SARS-Co-V will differentially bind to the OE resulting in localized inflammation.

As the potential for airborne transmission of SARS-CoV-2 becomes increasingly accepted by the medical community [6, 7], clues derived from olfactory physiology, objective smell testing, and imaging studies converge around airborne aerosol exposure as an explanation for the widespread, profound, and relatively isolated OD in COVID-19. Our study is limited by a small number of subjects as well as uniformity of ambient conditions; temperature and humidity changes in alternate conditions could also affect intranasal diffusion, sedimentation, or other variables contributing to deposition of particulate. However, in this small number of subjects our data experimentally shows that despite bypassing the majority of the upper airway, smaller airborne aerosols appear to differentially deposit in significant concentrations within the olfactory epithelium. This provides a compelling aerodynamic mechanism to explain the common and relatively isolated olfactory dysfunction associated with the majority of COVID-19 infections.

## Supporting information

**S1 Dataset. Raw data of maximum value, mean gray value of non-zero pixels, and standard error of non-zero pixels for each subject and condition.**
(XLTX)

## Author Contributions

**Conceptualization:** Alan D. Workman, Aria Jafari, Benjamin S. Bleier.

**Data curation:** Alan D. Workman, Aria Jafari, Roy Xiao.

**Formal analysis:** Alan D. Workman, Roy Xiao.

**Investigation:** Alan D. Workman, Aria Jafari, Roy Xiao, Benjamin S. Bleier.

**Methodology:** Alan D. Workman, Aria Jafari, Benjamin S. Bleier.

**Project administration:** Benjamin S. Bleier.

**Resources:** Benjamin S. Bleier.

**Supervision:** Aria Jafari, Benjamin S. Bleier.

**Writing – original draft:** Alan D. Workman, Roy Xiao, Benjamin S. Bleier.

**Writing – review & editing:** Aria Jafari, Benjamin S. Bleier.

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
