## [Decision Letter · Decision Letter 0]

27 Oct 2020

PONE-D-20-22099

Airborne Aerosol Olfactory Dispersion Contributes to Anosmia in COVID-19

PLOS ONE

Dear Dr. Workman,

Thank you for submitting your manuscript to PLOS ONE. After careful consideration, we feel that it has merit but does not fully meet PLOS ONE’s publication criteria as it currently stands. Therefore, we invite you to submit a revised version of the manuscript that addresses the points raised during the review process.

We look forward to receiving your revised manuscript.

Kind regards,

Vladimir Mikheev

Academic Editor

PLOS ONE

Journal Requirements:

2. In your methods section and your ethics statement in the online submission form please clarify whether consent was informed consent.

3. Please provide the source and catalog number for the FUL-GLO Fluorescein Sodium used in this study.

4. To comply with PLOS ONE submission guidelines, in your Methods section, please provide additional information regarding your statistical analyses.

For more information on PLOS ONE's expectations for statistical reporting, please see https://journals.plos.org/plosone/s/submission-guidelines.#loc-statistical-reporting

5. Please include captions for your Supporting Information files at the end of your manuscript, and update any in-text citations to match accordingly. Please see our Supporting Information guidelines for more information: http://journals.plos.org/plosone/s/supporting-information

Additional Editor Comments:

Dear Dr. Workman,

Deep apologies for this long delay.

It was really hard to find reviewers but now we have necessary minimum number of reviews.

I think this paper is important one but you need to address minor comments suggested by the reviewers.

Again sorry for the delay.

Reviewers' comments:

Reviewer's Responses to Questions

**Comments to the Author**

1. Is the manuscript technically sound, and do the data support the conclusions?

Reviewer #1: Yes

Reviewer #2: Yes

2. Has the statistical analysis been performed appropriately and rigorously? 

Reviewer #1: I Don't Know

Reviewer #2: Yes

3. Have the authors made all data underlying the findings in their manuscript fully available?

Reviewer #1: Yes

Reviewer #2: Yes

4. Is the manuscript presented in an intelligible fashion and written in standard English?

Reviewer #1: Yes

Reviewer #2: Yes

5. Review Comments to the Author

Reviewer #1: The objective of this study is to compare aerosol deposition in the OE and RE of nasal passages. Measurements clearly show that small particles are able to reach the olfactory region (mainly by diffusion given that there is very little airflow into the region) while large particles lack diffusive properties and hence travel mainly through the RE as evidenced by its deposition in this region. COVID particles are considered fine and thus can penetrate the OE region. This reviewer has the following comments regarding the manuscript:

The Method section is confusing and lacks details. It needs to clearly describe the experimental system for each particle type, choice and reason for the selection of two classes of particles (lines 61 – 63). Droplets are aerosols too. You need to distinguish the aerosols by size and composition. Also, need to explain why you picked large particles versus small.

Lines 44 – 46: The statement is partly inaccurate and the process is well described in the aerosol literature. The inhaled air splits at the junction with the majority of the flow (and thus particles regardless of the size) pass through the RE region. However, there is little deposition due to high flow convection in the RE region (thus small gravitational settling). There are far fewer particles reaching the OE region. However, those that reach there deposit by Brownian diffusion and sedimentation. The reason for higher deposition in OE is not due to having more particles but having more deposition. Please revise your hypothesis and clearly explain how you are going to achieve your hypothesis.

Line 60: you did not validate your size distribution but measured it.

Reviewer #2: The authors should be commended for this study, which hypothesizes that an aerodynamic mechanism may help explain the link between COVID-19 and olfactory dysfunction. Specifically, they show that airborne aerosols differentially deposit in significant concentrations within the olfactory epithelium. This paper, which is brief but methodologically sound, represents a valuable addition to the literature.

I would suggest that the authors address several issues prior to potential publication. A primary concern is that there is no mention in the methods/results section of how many subjects were analyzed. The authors should be more clear that only 3 patients (based on the raw data) were analyzed. This small number limits the conclusions one can make from this data and this should be explicitly stated in the discussion section. Furthermore, there is no descriptions of the subjects other than 'healthy.' Further limitations should be added to the discussion section.

6. PLOS authors have the option to publish the peer review history of their article (what does this mean?). If published, this will include your full peer review and any attached files.

Reviewer #1: **Yes: **Bahman Asgharian

Reviewer #2: No

---

## [Author Response · Author response to Decision Letter 0]

24 Nov 2020

Dear Editors:

Thank you very much for your consideration and response, as well as for the helpful comments of the reviewers. We have made changes to the manuscript in line with the requests. These are outlined below and are highlighted in the manuscript text. We are confident that the revised work will be of interest to the PLOS ONE readers.

Best regards,

Alan Workman, MD, MTR

Benjamin Bleier, MD, FACS, FARS

Associate Editor Comments

 These style requirements have now been added to the updated manuscript. 

2. In your methods section and your ethics statement in the online submission form please clarify whether consent was informed consent.

The statement, “…and informed consent was obtained from all participants was added to the first line of the methods.

3. Please provide the source and catalog number for the FUL-GLO Fluorescein Sodium used in this study.

The catalog number is now included. The source was Akorn Inc., Lake Forest, IL, USA (written in text as well).

4. To comply with PLOS ONE submission guidelines, in your Methods section, please provide additional information regarding your statistical analyses.

A separate section on statistical analysis has been added to the methods, with enough detail that utilizing our raw data as well as the statistical tests described our results would be easily reproducible. 

5. Please include captions for your Supporting Information files at the end of your manuscript, and update any in-text citations to match accordingly. 

The file has been renamed and the captions have been added. 

Reviewer 1 Comments

1. The Method section is confusing and lacks details. It needs to clearly describe the experimental system for each particle type, choice and reason for the selection of two classes of particles (lines 61 – 63). Droplets are aerosols too. You need to distinguish the aerosols by size and composition. Also, need to explain why you picked large particles versus small. Lines 44 – 46: The statement is partly inaccurate and the process is well described in the aerosol literature. The inhaled air splits at the junction with the majority of the flow (and thus particles regardless of the size) pass through the RE region. However, there is little deposition due to high flow convection in the RE region (thus small gravitational settling). There are far fewer particles reaching the OE region. However, those that reach there deposit by Brownian diffusion and sedimentation. The reason for higher deposition in OE is not due to having more particles but having more deposition. Please revise your hypothesis and clearly explain how you are going to achieve your hypothesis.

The hypothesis has now been revised to be significantly more clear throughout all sections of the manuscript. As you discussed, we have now distinguished between “persistently aerosolized particulate” of smaller size (sub 5-10�m) and larger aerosolized droplets (30-100�m), with the former generated by the nebulizer and the latter generated by the atomizer. This distinction between the size of particulate forms the basis of our hypothesis, in that the smaller particles will not deposit in the RE but do deposit in the OE by diffusion, while larger particulate does indeed settle in the RE. The composition of the particulate remains the same (identical solutions). We have additionally changed the text to highlight that our hypothesis is that small airborne aerosols selectively deposit in the OE, not that disperse to the OE. 

2. Line 60: you did not validate your size distribution but measured it.

 This has been corrected to utilize the word “measure.”

Reviewer Two Comments

1. A primary concern is that there is no mention in the methods/results section of how many subjects were analyzed. The authors should be more clear that only 3 patients (based on the raw data) were analyzed. This small number limits the conclusions one can make from this data and this should be explicitly stated in the discussion section. Furthermore, there is no descriptions of the subjects other than 'healthy.' Further limitations should be added to the discussion section.

We have now listed “n=3 subjects” in the description of of the raw data and statistics within the results section. Within the methods, we have added the sentences “Participants were healthy subjects between the ages of 25 and 35 without a history of chronic rhinosinusitis, allergic rhinitis, or other rhinologic disease. At the time of data collection, subjects did not have any symptoms of acute sinusitis, rhinorrhea, or subjective nasal obstruction.” In the discussion section, we have now noted additional limitations, and have added the sentence “our study is limited by a small number of subjects as well as uniformity of ambient conditions; temperature and humidity changes in alternate conditions could also affect intranasal diffusion, sedimentation, or other variables contributing to deposition of particulate.”

---

## [Editor Report · Decision Letter 1]

1 Dec 2020

PONE-D-20-22099R1

Airborne Aerosol Olfactory Deposition Contributes to Anosmia in COVID-19

PLOS ONE

Dear Dr. Workman,

Thank you for submitting your manuscript to PLOS ONE. After careful consideration, we feel that it has merit but does not fully meet PLOS ONE’s publication criteria as it currently stands. Therefore, we invite you to submit a revised version of the manuscript that addresses the points raised during the review process.

Dear Authors,

Thank you for revision of your manuscript.

I believe you properly addressed Reviewers comments.

I think your manuscript is almost ready for publication but I would like you to address couple of my comments as well.

Reviewer 1 stated: "There are far fewer particles reaching the OE region. However, those that reach there deposit by Brownian diffusion and sedimentation."

Please note that although in general its true (both Brownian diffusion and sedimentation play role) but talking specifically about COVID-19 it has to be noted that single COVID-19 particles (containing only one virion) could be of ~100 nm (60-150 nm) size but peak concentrations found in hospitals were within ~250-500 nm size (please see reference below). Particles of this size range should not deposit by Brownian diffusion (rather particles below 100 nm are subject to Brownian diffusion).

Therefore I would strongly suggest to edit your statement made on L56-57 (Caption to Figure 1) "proportion of these aerosols diffusing through Brownian motion into the olfactory epithelium (OE)" to "proportion of these aerosols penetrating into the olfactory epithelium (OE)".

Also, L67 - please specify what type of nebulizer was used to generate smaller sized particulate in the sub 5 micron range.

Reference:

Liu Y, Ning Z, Chen Y, Guo M, Liu Y, Gali NK, et al. Aerodynamic analysis of 193 SARS-CoV-2 in two Wuhan hospitals. Nature. 2020; https://doi.org/10.1038/s41586-194 020-2271-3

We look forward to receiving your revised manuscript.

Kind regards,

Vladimir Mikheev

Academic Editor

PLOS ONE

---

## [Author Response · Author response to Decision Letter 1]

3 Dec 2020

Dr. Mikheev,

Thank you again for your ongoing consideration of our manuscript. We are happy to address your comments and agree with you that the requested changes make the manuscript clearer and are more harmonious with the existing literature on SARS-CoV-2. The changes have been made in the text (line 56-57 for the first comment) and addition of the nebulizer model to line 67. 

Best regards,

Alan Workman, MD, MTR

Benjamin Bleier, MD, FACS, FARS

---

## [Editor Report · Decision Letter 2]

4 Dec 2020

Airborne Aerosol Olfactory Deposition Contributes to Anosmia in COVID-19

PONE-D-20-22099R2

Dear Dr. Workman,

We’re pleased to inform you that your manuscript has been judged scientifically suitable for publication and will be formally accepted for publication once it meets all outstanding technical requirements.

Kind regards,

Vladimir Mikheev

Academic Editor

PLOS ONE
---

## [Editor Report · Acceptance letter]

27 Jan 2021

PONE-D-20-22099R2 

Airborne Aerosol Olfactory Deposition Contributes to Anosmia in COVID-19 

Dear Dr. Workman:

I'm pleased to inform you that your manuscript has been deemed suitable for publication in PLOS ONE. Congratulations! Your manuscript is now with our production department. 

Kind regards, 

on behalf of

Dr. Vladimir Mikheev 

Academic Editor

PLOS ONE